# The Winner Takes it All: Risk Factors and Bayesian Modelling of the Probability of Success in Escaping from Big Cat Predation

**DOI:** 10.3390/ani12010051

**Published:** 2021-12-28

**Authors:** Sergio Fernández Moya, Carlos Iglesias Pastrana, Carmen Marín Navas, María Josefa Ruíz Aguilera, Juan Vicente Delgado Bermejo, Francisco Javier Navas González

**Affiliations:** 1Department of Genetics, Faculty of Veterinary Sciences, University of Córdoba, 14014 Cordoba, Spain; sergiofm1995@hotmail.es (S.F.M.); ciglesiaspastrana@gmail.com (C.I.P.); carmen95_mn@hotmail.com (C.M.N.); juanviagr218@gmail.com (J.V.D.B.); 2Department of Conservation, Córdoba Zoo Park, 14004 Cordoba, Spain; conservador.zoo@ayuncordoba.es; 3Department of Agriculture and Ecological Husbandry, Area of Agriculture and Environment, Andalusian Institute of Agricultural and Fisheries Research and Training (IFAPA), Alameda del Obispo, 14004 Cordoba, Spain

**Keywords:** felines, predator, prey, risk factors, Bayesian regression modelling

## Abstract

**Simple Summary:**

Predation is a complex behavioural interaction that is conditioned by biotic and abiotic factors. In their struggle for survival, the agents participating in the hunt interaction adapt their strategies seeking an opposite interest which leads to the same outcome, success in surviving. Predator/prey interaction data was obtained from on-line posted videos. The examination of records suggested that the species and age range of the predator, its status at the end of the hunt, the time elapsed between the sighting of its prey and the physical contact with it, the species of the prey and the relief of the land were determined success of escape of the prey in case of attack. The present study sheds light on the multietiological nature of predatory abilities and the strategies to fend off anti-predation strategies of the prey in big cats. The theoretical and empirical contents derived from this work will allow the design of environmental enrichment programs in captivity to be substantially improved by providing preys and enough space for them to express big cats predating strategies. The extrapolation of these results to domestic contexts may enable approaching selection strategies from two perspectives, with the aim to boost predating ability of domestic felids for pest control or to enhance defence in domestic ruminant prey from big cats.

**Abstract:**

The individuals engaged in predation interactions modify their adaptation strategies to improve their efficiency to reach success in the fight for survival. This success is linked to either capturing prey (predator) or escaping (prey). Based on the graphic material available on digital platforms both of public and private access, this research aimed to evaluate the influence of those animal- and environment-dependent factors affecting the probability of successful escape of prey species in case of attack by big cats. Bayesian predictive analysis was performed to evaluate the outcomes derived from such factor combinations on the probability of successful escape. Predator species, age, status at the end of the hunting act, time lapse between first attention towards potential prey and first physical contact, prey species and the relief of the terrain, significantly conditioned (*p* < 0.05) escape success. Social cooperation in hunting may be more important in certain settings and for certain prey species than others. The most parsimonious model explained 36.5% of the variability in escaping success. These results can be useful to design translatable selective strategies not only seeking to boost predation abilities of domestic felids for pest control, but also, biological antipredator defence in potential domestic prey of big cats.

## 1. Introduction

Predation can be defined as the biological relationship between two species in which one of them (predator) seeks to hunt the other (prey) for subsistence, as this is their only vital feeding strategy [1]. In most cases, predator and prey belong to different species that are immersed in a complex cycle of interdependent relationships within the pyramidal hierarchy of the food chain [2]. However, alternatively, predators can be, with relative frequency, another organism’s prey, and likewise prey are often predators. This means, a given predator can also be prey for other species with which it shares habitat and competes for trophic resources present in it, just as a species becomes prey for different predators [3]. Consequently, in the broad sense, predatory behaviours involve those patterns by means of which predators, in any of the aforementioned cases, usually motivated by hunger, but not always, kill and eat prey [4].

Among predators there is a large degree of specialization. In this regard, ecological theory classifies predatory species into two main groups according to the size of the ecological niche that they represent: reduced trophic niche or low prey variety in specialist predators (e.g., felids such as the Iberian lynx (*Lynx pardinus*), birds of prey such as the Barn Owl (*Tyto alba*) or mustelids such as European polecats (*Mustela putorius* L.) among others) and broad trophic niche or varied prey in generalist predators (spotted hyena (*Crocuta crocuta*)*,* African leopard (*Panthera pardus pardus*) or bald eagle (*Haliaeetus leucocephalus*)*,* among others) [5]. A third type of predator is the so-called facultative specialist, because of its rigorous capacity to adapt its predatory strategy to specific and generally transitory environmental conditions [6].

Additionally, many predators have exogenous rhythms of predatory behaviour which may relate to the availability of prey among others, or endogenous rhythms which may persist in independence of environmental factors [7]. In these contexts, predatory behaviours may be adapted to the circadian routines that predators and preys present with some species being diurnal while others are strictly nocturnal or may feed/chase both during day and night [8].

Schematically, the act of predation consists of a maximum of four stages: prey detection, attack, capture and consumption [9]. The relationship between predator and prey is one which typically benefits the predator, in detriment of the prey species. Predatory behaviours can be classified depending on trophic level or diet, specialization, or the nature of the predator’s interaction with prey. In this regard, classification of predators by the extent to which they feed on and interact with their prey consider two factors: physical closeness between the predator and prey/host and, whether or not the prey are directly killed by the predator.

By virtue of its ethological nature, predatory behaviour not only contributes to the adaptation and survival of animal species, but also appears as a specific pattern of transgenerational transmission behaviour. Such a pattern is necessary to guarantee an adequate status of animal welfare in the species involved, both in their natural environment and in captivity [10]. Sometimes, predation has even been reported to cause indirect benefits to prey species [11], though the individuals preyed upon themselves do not benefit [12]. This means that, at each applicable stage, predator and prey species are in an evolutionary arms race to maximize their respective abilities to obtain food or avoid being eaten. Contextually, these animals play a fundamental role in maintaining the balance of natural ecosystems as regulatory factors of the population dynamics of wild prey [13]. That is, they regulate, restoring the balance in the prey population, or limit, establishing the equilibrium density of the prey, their populations in quantitative terms [14]. The resulting effects on wild prey populations, both positive and negative, will depend to a large extent on the biology and abundance of the prey and predators, as well as the trophic ecology of the latter [14,15,16].

Indeed, selective pressures imposed on one another may often lead to such an evolutionary arms race between prey and predator, resulting in various antipredator adaptations in both groups, which may reflect changes at a genetic level.

In this sense, five types of predatory behaviour have been described as follows: true predation, grazing, parasitism, parasitoidism and competitive predation. True predation is the main strategy implemented by big cats and defines the predatory pattern in which a true predator can commonly be known as one which kills and eats another organism. Dawkins and Davis [17] offered an alternative view of predation as a form of competition within which intraguild predation is found. Intraguild predators are those that kill and eat other predators of different species at the same trophic level, and thus that are potential competitors, as happens among big cats sharing the same niche [18].

For instance, recent studies have shown that large predators, in addition to affecting and regulating the populations of their prey, also do so on those carnivorous species that are below them in the food chain, the so-called mesopredators [19,20,21]. When a super predator disappears or its population is reduced, mesopredators rapidly increase their reproduction (Hypothesis of “Release of Mesopredators”) [22], and being more generalist and opportunistic, they can drive prey species to extinction [23,24]. In short, the presence and balance maintained by super predators allows for the conservation and survival of prey species and the preservation of the natural environment.

Predation has always been one of the cornerstones in the theories of evolution as the species involved undergo a series of physical and behavioural adaptations that help them in this struggle for survival [25]. While feline predators generally develop claws, speed, sharp teeth, agility, and the ability to organize themselves into a group to hunt or attack their prey unexpectedly, the latter acquire various skills that allow them to flee from their predators (speed, camouflage, or conditioned aversion by releasing chemical signals that are unpleasant to the taste or smell of their predator) [26,27].

Big cats, one of the groups of predators that dominate a large part of the world’s ecosystems, have managed to evolve different techniques to bring down their prey, which has provided them with their great ecological success [28]. Within this taxonomic group, we strictly include those cats that roar, that is, the four species of the genus *Panthera* (lion, leopard, jaguar and tiger).

Lions (*Panthera leo*), cats with a markedly gregarious character, have aroused the interest of scientific groups investigating their strategies and success in hunting, in which females are the main protagonists [28,29,30,31,32,33]. These tend to organize themselves into groups that pursue their prey (medium-sized mammals, mainly ungulates), while males take an active part in hunting actions when they do not belong to any defined group of congeners or in cases where the prey is of considerable size (buffalo, hippopotamus or elephants, for example) as success lies in the coordinated cooperation of several group members [34].

By contrast, the leopard (*Panthera pardus*) is one of the most versatile species since it is adapted to multiple habitats on the African and Asian continents. For this reason, it is one of the big cats whose predatory behaviour is directed to a diverse cast of prey (small and medium-sized primates, fish, snakes and crocodiles). In addition to stalking them stealthily by taking advantage of the camouflage provided by the dense vegetation, their ability to jump and climb trees gives them a particular skill for hunting tree species and the opportunity to protect their food from other predators in open terrain [35].

The jaguar or *Panthera onca* presents a hunting strategy similar to that of the leopard and is one of the species with the most powerful jaws in the entire animal kingdom [36,37]. It shows a great predilection for aquatic terrain, so among its frequent prey are turtles and crocodiles as well as large land mammals such as deer or capybaras. As for their hunting technique, they prefer to deliver a deadly blow from their hiding place rather than to chase their prey and usually kill them by breaking their skull instead of asphyxiating them as other cats do [36].

The tiger (*Panthera tigris*) inhabits different ecosystem complexes, and its favourite prey are large ungulates. They prefer to use their strength to knock down their prey with one blow and can reach up to 90 km/h during the chase. This big cat is also characterized by its ability to jump with strength and power over its prey in order to knock it down and immobilise it [38,39].

The cheetah (*Acinonyx jubatus*) is one of the cats with the highest visual acuity. When it comes to hunting, instead of rushing at its prey, it prefers to approach it with caution and to start the chase only when the possibilities of success are perceived to be high. Depending on the terrain, this felid can adopt different tactics: if it finds it easy to approach its prey silently, it will proceed to attack its prey from the closest possible location, while if it finds it difficult to observe from a distance, it will approach it in a race from the longest distance [40,41].

The puma or mountain lion (*Puma concolor*) ordinarily lives and hunts alone using its powerful leap and short run to bring down its prey [42,43]. On the other hand, the snow leopard (*Panthera uncia*)*,* has a relative predilection for ambush and then chasing its prey over the rocks and grabbing it by the neck with its teeth [44,45].

Numerous factors influence successful predation in big cats. In very wide spaces and with scarce vegetation cover, the hunt-in-group approach becomes an adaptive strategy to compensate for the inconveniences derived from the limitations of camouflage at the time of stalking its prey. The time of day and the lunar phase are also cited as determining factors that influence the feeding behaviour of these animals [46,47]. The species, group size and anti-predatory strategies of the prey, as well as the size and composition of the group and the hunting techniques employed by the «cat herd or pride» (e.g., Lion), will ultimately determine the success or failure of the hunting activities [48,49].

The aim of this work is the qualitative and quantitative study of the factors that affect the escape success of prey from hunt interactions in big cats. Through the evaluation of day and night video footage of predator/prey interactions obtained from various web sources, we designed a predictive model that allows determining the impact of combinations of these factors on the escape success. The theoretical knowledge of these conditioning elements in the predatory behaviour of big cats will allow, on the one hand, the reinforced approach of selection strategies potentially extrapolated to domestic animals, while it will seek to promote the conservation and transmissibility of desirable behaviour patterns in domestic feline species that can act as predators (for example, the cat) or as prey (conflicts between domestic livestock and wildlife, for example). Secondly, the design of environmental enrichment plans for animals in captivity may be strengthened, thus allowing a substantial improvement in the general welfare of the animals.

## 2. Materials and Methods

The methodology used in this work is based on the work of Kydd et al. [50], who extracted animal research data from videos available on the YouTube digital platform, as an attempt to promote research from pre-existing resources.

### 2.1. Total Sample and Sources

Complete sample comprised a total of 308 videos (clips and/or documentaries), hosted on public (YouTube, www.youtube.com and private (National Geographic, www.nationalgeographic.com and DisneyPlus, www.disneyplus.com, accessed on 16 May 2020) digital platforms. These videos contained footage about the predation behaviour of the main big cat species (lion, leopard, jaguar, tiger, cougar, cheetah, snow leopard, panther, serval, ocelot and caracal) [51,52].

The search for the graphic material hosted in YouTube digital platform was performed using the keywords “big cats”, “big cat hunting”, “big cat success in hunting”, “big cat fail in hunting”, “lion hunting”, “leopard hunting” were used, “jaguar hunting”, “tiger hunting”, “cougar hunting”, “cheetah hunting”, “snow leopard hunting”, “melanistic leopard/panther hunting”, “serval hunting”, “ocelot hunting”, “caracal hunting” and their respective homonyms in Spanish. In the case of National Geographic and DisneyPlus, those specific documentaries on biology and ecology in big cats were directly selected.

### 2.2. Study Sample

Due to the nature of the study and the sources from which information was taken, only those videos on which a continuous hunting attempt had been displayed were considered. This criterion was followed to prevent data distortion and biases derived from the splicing process of editing unrelated footage together to garner a logical narrative. 

Once selected, videos whose sharpness and total duration did not allow the complete analysis of the predation behaviour (sighting, stalking, chasing and attack) and the defence strategies of the prey from its beginning to its conclusion, were discarded (*n* = 64). As a result, the study sample of this research work comprised a total of 244 videos: 80 (32.80%) videos of predation behaviour in lions, 45 (18.50%) of leopards, 11 (4.50%) of jaguars, 40 (16.40%) of tigers, 13 (5.30%) of pumas, 43 (17.60%) of cheetahs, 7 (2.90%) of snow leopards, 2 (0.80%) of panthers and 1 (0.40%) for each of the three smaller big cat species (serval, ocelot and caracal). The videos were analysed between 18 February and 15 May 2020.

### 2.3. Factor and Variable Recording

The visualization of selected videos permitted the registration of the different qualitative factors and quantitative variables in regards the climatic and environmental characteristics and behavioural patterns of the predator and prey involved displayed in each recorded hunt event. Certain factors, such as the flight distance from prey to predator [53] could not reliably be measured, hence their recording was unfeasible. All the information extracted from the videos was organized as shown in Table 1.

### 2.4. Preliminary Testing of Statistical Properties

Firstly, the assumptions of normality, homoscedasticity and multicollinearity were tested to determine the most appropriate statistical approach to follow. The normality testing routine used was Shapiro-France (for samples 5 ≤ *n* ≤ 1000) from the Stata Distribution Graphics Software Version 15.0 (StataCorp LLC, College Station, TX, USA). Levene’s test was used to evaluate the homogeneity of variance (homoscedasticity) between groups of different independent variables using SPSS Statistics for Windows software, Version 25.0 (IBM Corp., Armonk, NY, USA).

The preliminary evaluation of the sample properties revealed that the parametric assumptions had been violated, therefore the possibility of performing a parametric approach for data analysis was discarded. In the case of relatively small samples, the probability of finding significant results decreases [54]. Sample size limitation often translates into an additional difficulty in finding a biological relationship that can serve as a basis for explaining and interpreting the results obtained (Button et al., 2013). In this context, since Bayesian analyses are a plausible alternative given smaller data sets can be evaluated without the effects of loss of test power or precision conditioning the results, as suggested by Hox et al. [55] and Lee and Song [56]. Specifically, these authors suggested that Bayesian estimation methods require a much lower ratio between measured parameters and observations (1:3 instead of 1:5).

In our case, a Bayesian inference for ANOVA was performed to detect the existence of differences in the mean for the probability of escape success (dependent variable) between the different levels of the independent variables (predator species, sex of the predator, age range of the predator, type of attack, hunting mode, number of predators, time from first attention to action, time from first attention to interaction or direct contact with the prey, point of grasp of the predator on its prey, state of the predator at the end of the hunt, hunting attempts, species of the prey, number of prey, sex of the prey, age range of the prey, time of day, atmospheric conditions and relief).

As suggested in the document IBM SPSS Statistics Algorithms version 25.0 by IBM Corp (IBM Corp., Armonk, NY, USA). [57], the Bayesian inference for ANOVA is approached as a special case of a multiple linear regression model. A complete description of the algorithms used by SPSS software to develop the test described in this study can be found in the previously mentioned public document. The tolerance value for the numerical methods and the number of method interactions were determined by default by IBM SPSS Statistics, version 25.0 [58].

The list of criteria proposed by Depaoli and Van de Schoot [59] was used to detect (a) possible incidences before estimating the model, (b) after estimating the model but before interpreting the results, (c) to understand the influence of the priors, and (d) to determine the actions to be taken after interpreting the results.

Effects of the factors considered in the predictive models are quantified by evaluating their confidence interval in the posterior distribution statistics. Third, the 95% Credibility Interval shows that there is a 95% probability that these regression coefficients (Posterior distribution true value for each covariate and factor) in the population lie within the corresponding intervals. When 0 is not contained in the Credibility Interval, the effect is evidently not 0 and a significant effect for such factor is detected. Bayes factor (BF) was calculated to determine the probability of both the null hypothesis and the alternative or one model versus the other based on the *a priori* distribution of the data. This factor quantifies the change in probability from the *a priori* distribution to the posterior or a posteriori distribution due to the data. The Bayes factor (BF) was then calculated to determine the validity of the model containing the significant factors and covariates compared to a model that only considers the intercept. The BF is a measure of the strength of evidence and is used instead of *p*-values (frequency approximations) to draw conclusions. A large BF implies that the evidence favours the alternative hypothesis compared to the null hypothesis. The levels commonly used to define evidence significance levels are established following the premises of Jeffreys [60] and Lee and Wagenmakers [61]. In this context, Cleophas and Zwinderman [62], suggested a method to extrapolate between the Bayes factor used in Bayesian approaches and *p* values from frequentist approaches to favour the interpretability of results.

### 2.5. Bayesian Approach to Linear Regression Analysis

A predictive model for the probability of escape success as a dependent variable was designed considering the combination of factors and covariates for which significant effects had been reported on the probability of escape success among all those listed in Appendix A. The regression model followed the following general equation:y_n_ = X_1_β_1_ +… X_n_β_n_ + ε_n_(1)
where n = 1, 2, …i is the nth number of factors/covariates, and n is the vector of observations for the escape probability variable with a dimension of n (244 observations); X_n_ is the incidence matrix for each of the factors in the model, and β_n_ the standardized regression coefficients for the nth number of factors and covariates considered for the model that as we have mentioned before have been described previously in Table 1.

The intercept or constant (μ, mean of the response when all predictors are 0) was forced to be 0 (crossing the origin), as suggested by Brewer [63] since the model designed in the present work is simple and the coefficients had been standardized. Additionally, Brewer (2002) indicated that precautions should be taken before using an intercept other than 0 in the model unless there is no empirical need (for example, when using non-standardized coefficients). In fact, the estimated confidence intervals for the estimated intercept were used as empirical indicators for the need for an intercept. The distribution followed by the residual effects (ε_n_) was assumed to be normal and defined by $\varepsilon_{i}|X_{i}N\left(0,\;\sigma_{\varepsilon i}^{2}\right),$ where *X_εi_* is an identity matrix and $\sigma_{\varepsilon i}^{2}$ is residual variance, respectively. The Bayesian approach to regression analysis developed in this study was carried out with the SPSS Statistics for Windows, version 25.0, IBM Corp. (2017) [58]. In addition, the linear Bayesian regression routine in Stata software version 16.0 [64] was used to calculate descriptive statistics of the posterior distribution of each factor and independent covariate included in the model for the variable dependent on the probability of successful dam escape. A full description of the algorithms used by SPSS to perform the Bayesian approach to linear regression analysis can be found in the public document IBM SPSS Statistics Algorithms v. 25.0. by IBM Corp [57].

For the analyses described in this study, the Jeffrey–Zellner–Siow g-prior mix was used [65]. The Jeffrey–Zellner–Siow priors depend on the variable X_n_ and, therefore, on the data in the study; although this fact does not suppose a setback since strictly the regression models depend equally on X_n_.

## 3. Results

Table 1, Table 2 and Table 3 show a summary of the results of the Bayesian two-tailed ANOVA approach to detect differences in the mean probability of successful prey escape between the levels of different factors and covariates related to predator, prey, and environmental conditions existing at the time of the hunt. As can be observed, the factors ‘Predator species’, ‘Predator age range’, ‘Predator status at the end of the hunt’, ‘Prey species’ and ‘Relief ‘, were variously responsible for significant differences in the mean for the probability of successful escape of the prey (*p* < 0.05).

Results of the frequency analyses of the causes for escape success can be visualised in Figure 1. The defensive mechanisms implemented by prey (either it is physical features, back attacking, flighting or counterpart protection) are the most frequent causes for escape success rather than attacking fails. Frequencies of successful escapes when this factor was considered ranged from over 10% to 23.60%. However, habitat characteristics still represent the main cause for escape success in 12.70% of the times.

Table 4, Table 5, Table 6, Table 7 and Table 8 show a summary of the descriptive statistics of the posterior distribution for those factors for which significant differences in mean probability of escape success were found were found (‘Predator species’, ‘Predator age’, ‘Prey species’, ‘Status of the predator at the end of the hunt’ and ‘Relief’).

Table 9 reports that the time between the predator’s first attention to its prey and direct contact with it in those cases where the hunting scene was successful (the prey did not escape), was on average 39.043 s, while when the capture was unsuccessful (the prey managed to escape) this time was highly significantly (*p* < 0.01) higher (74.333 s on average).

Table 10 and Table 11 report the summary results for the Bayesian ANOVA inference for the model predicting the probability of prey’s successful escape including the factors for which significant effects had been detected. The proposed combination of factors explains a 36.5% of the variability in the probability of escape success and the model was found to be significantly more likely (F:1.435, *p* < 0.05) than one that considered only the intercept, i.e., that did not consider any of the recorded factors.

## 4. Discussion

Bayesian inference maximizes the ability to detect significant effects in limited sample size contexts. Consequently, a much smaller ratio of parameters to observations is required (1:3 instead of 1:5) [66]. As a result, potentially distorting conditions derived from sampling, for instance, unequal numbers of observations across group members (i.e., predators), can be sorted. Still, these sample limitations often reflect in the broadening of confidence intervals (as error term increases), which makes it compulsory for results to be supported on acceptable values of Bayes factor.

For this, certain cautions must be taken. In the specific case of linear regression, the choice for Jeffrey–Zellner–Siow (JZS) prior is especially appropriate, given this prior is symmetric, centred at zero and scale invariant. This means positive and negative values of the slope parameters have a priori the same probability to occur and implies that Bayes factor values do not change if variables, factors or covariates, measured in different units, are evaluated together, as common in multifactorial field studies [66].

Our study reported inherent and environmental factors that can condition the fate of a specific hunting event. Among these factors, the unique characteristics of each big cat species particularly shape the different hunting techniques that they implement, with the main aim of obtaining a successful outcome from a hunt interaction.

### 4.1. Predator-Dependent Factors

Predation is an etiologically complex process whose success depends on multiple inherent and external factors that determine the success of hunting stages [67,68]. One of the main factors conditioning predation success is predator species. Our study suggested tigers to be the most successful hunters on average, followed by the leopard, jaguar, cheetah and lion with the snow leopard and puma being the least likely cat species to succeed in hunting. Contextually, most predators selectively stop hunting and/or eliminate certain prey categories under certain conditions, previously identified by olfactory/visual stimuli and stalked by complex behavioural patterns which may differ across species [69].

Predation success increases as predators age, with older animals being more likely to capture prey than younger animals. According to literature, adult individuals are more effective in stalking prey with a successful outcome, probably due to the experience achieved as the animal matures [68]. Alternatively, an advanced biological age will imply, in general terms, a greater size and body development. This has been suggested to enable predators to approach prey of progressively considerably higher sizes with time [70,71,72]. By contrast, no significant differences were reported when the sex of the predator was considered, contrary to other studies reporting a relatively greater probability of success in males than females in some feline predators, such as lions [48].

Closely related to the above, the general status of the predator at the end of the hunt (healthy, injured or dead) significantly explained the success of the escape of the prey. Age, practical experience and predator general status of health, could be determining provided the efforts that individuals dedicate to stalking, persecution and sacrifice of its prey may be enhanced in time [68]. In this sense, adult animals, with a general status of good health and with outstanding experience in the hunting tasks, are expected to be more efficient in the successful conclusion of its predatory behaviour without this repressing in notable physical injuries derived from the defensive strategies exerted by the prey.

Social cooperation in the hunting tasks has also been reported to be a significant factor in the achievement of success [73,74]. In these regards, a very common strategy in lions, but which has not been reported in other feline species, consists of all the members of the group contributing in a coordinated way to the attempts of stalking prey [73]. Consequently, the probability of capturing larger animals such as giraffes (*Giraffa camelopardalis*)*,* buffalo (*Bubalus bubalis*) or elephant cubs (*Loxodonta africana*) is substantially increased [74]. However, this predation type can also generate competition between congeners for the captured food. Conversely, cats which hunt alone are more likely to eat what they capture, even if this type of hunting involves a greater expenditure of energy to catch the prey and a greater risk of the prey escaping. In our case, however, no significant effect on hunting attempts success was reported for the type of attack executed (solo vs. group).

Our results suggest the time elapsed between potential prey identification and the first physical contact between predator and prey doubled when prey managed to escape. In this regard, Funston et al. [48] suggests predators that initiate the pursuit of their prey immediately after detection, have a greater probability of success in their capture than those that execute a previous stalking, of variable duration, to the pursuit *per se*.

### 4.2. Prey-Dependent Factors

Numerous factors related to the prey affect its ability to escape from the attack of predators, such as the animal species, age, sex and, in the case of species with defined hierarchical structures, the size of the social group [70,75]. Our results suggest that, among the range of prey dependent factors, only the prey species proved to be an influential factor in its probability of escape. This could mainly be ascribed to prey size, with the success of the hunt being greater when the size of the prey does not significantly exceed that of their predators, which means medium or large sized prey have greater probability of survival against the attack of predators [68].

While predators seek to refine their attack strategies, the defence strategy responds to an increase in the efficiency of its ability to escape [67,76]. However, avoiding being captured must be weighed against the satisfaction of prey’s basic needs. For this, prey have developed adaptation strategies to the natural cycles of regeneration and specific space-time availability of their trophic resources which places them in a context of special vulnerability (large migrations and forced displacements due to natural causes) [77,78]. Indirectly, the efficiency in the anti-predation patterns (vigilance, intraspecific communication, agility and speed) of prey species [79,80], can be known from the joint evaluation of the strategies and hunting success of its predators [81,82,83,84].

Our findings reported the existence of significant differences in the probability of successful escape across prey species. The relevance of these findings lies in the fact that, even if not all prey-predator possibilities were tested (not all prey possibilities were hunted by all the predator species in this study), each prey was tested as one of the possibilities in the range of potential prey for each particular big cat species. This not only permitted ranking the prey species independently (when only one or a few prey-predator pair was considered), but also permitted ranking those prey species which had been targeted across different big cat species.

Prey species were classified into three groups. The first group comprised medium and large sized herbivores, most of them being ungulates such as wildebeest, zebra, deer or antelope, which live in social groups. Most of this group’s members would escape due to the rear location of predator’s point of attachment. This could be justified by the ability of these prey to fight back with their hind legs in the escape race, thus harming the predator and interrupting its trajectory. The second group would be formed by aquatic animals such as the crocodile (*Crocodylus* spp.), the hippopotamus (*Hippopotamus* spp.) and the giant otter (*Pteronura brasiliensis*). Their main defence and escape mechanism would be the environment in which they live, since the only one that is familiar to this environment when hunting is the jaguar. The third group would be formed by medium and large sized omnivores like the bear (*Ursus arctos*) and the warthog (*Phacochoerus africanus*) and their success in escape may be justified by their ability to defend themselves with their fangs and biting back attacks. Generally, the success of prey managing to escape may be ascribed to the attachment point of predator, with target being the neck of prey in most attacks.

Contextually, literature mainly ascribes prey preferences to the balance between prey vulnerability and availability. In these regards, Hornocker [85] reported cougars in Idaho hunted the same number of moose (in North America) or elks (in Eurasia) (*Alces alces*) as mule deer (*Odocoileus hemionus*), even if the latter were more abundant. Moose and deer were apparently more vulnerable during winter, when forced into a hunting ground presenting the most favourable conditions for puma, as suggested by pumas also feeding on small prey. For instance, pumas feed primarily on wild pigs (*Sus scrofa*)*,* white-tailed deer (*Odocoileus virginianus*) and raccoons (*Procyon lotor*) [86] in South Florida, while small prey are also important in the diet of southeastern Peruvian pumas [87].

Jaguars in southeastern Peru feed primarily on larger prey, including deer (*Cervus elaphus*), capybaras (*Hydrochoerus hydrochaeris*) and peccaries (*Pecari tajacu*). Contextually, although the jaguar hunts agoutis (*Dasyprocta punctata*)*,* pacas (*Cuniculus paca*)*,* deer and capybaras in proportion to their abundance, they killed peccaries more frequently than it could have been expected probably because they were more vulnerable [87]. This was also evidenced by jaguars in Belize mainly preying on small prey, particularly armadillos (*Dasypus novemcinctus*) which are probably the most vulnerable prey in the area, while they hunted larger prey such as collared anteaters (*Tamandua tetradactyla*)*,* roe deer (*Capreolus capreolus*) and peccaries, when in smaller numbers [88].

The tigers in Chitwan prey on sambar deer (*Rusa unicolor*) more often than expected, based on their availability. This suggests the preference for such a large deer may rather depend on its higher vulnerability when compared to other smaller more abundant deer such as chital (*Axis* axis) or Indian hog deer (*Hyelaphus porcinus*). This may support the fact that the lack of evidence of gaur (*Bos gaurus*) being preyed upon by tigers, may presumably address its larger size to be a determinant preventive condition, [89]. By contrast, although in Chitwan gaur can only be found in the hills, there is evidence of tigers occasionally killing gaurs due to their higher presence in Kanha National Park in India [33].

The buffalo is the most abundant mammal in Lake Manyara National Park (Tanzania), and constitutes 62% of African lions’ diet [81,90]. In all, 81% of prey buffaloes were adult males, since they often separate from herds and are apparently the most vulnerable. Across Serengeti, lions mainly fed on wildebeests (*Connochaetes taurinus*) and zebras (*Equus zebra*) when abundant during annual migration; however, other times, buffalo and topi (*Damalicus lunatus*) are the main prey [81]. The most abundant African prey are also generally large. Indeed, Eloff [91] found that where large prey are relatively scarce, such as the Kalahari Desert, small mammals and juveniles constitute more than 50% of lion’s diet.

Kalahari desert leopard coexists with lion by hunting smaller prey [91]. According to Hoppe-Dominik [92], Ivory Coast leopards also feed on small prey, mainly small bovids such as duikers (*Raphicerus campestris*) with almost 40% of prey being arboreal (including seven primate species) [93]. Studies of leopard feeding habits in Rhodesia [94], Kenya [95], and northern Serengeti [96] also indicate that leopards’ main prey are small animals. However, in other areas, such as Kruger and Wilpattu Parks, they regularly hunt slightly larger prey [81]. Their diet appears to be the most varied of all big cats due to the diverse geographic locations in which they live. Although larger cats can subsist and reproduce upon smaller prey [88], they are morphologically specialized to kill as large as or larger prey than them. In this sense, big cats feed optimally with differences in feeding habits within a species largely reflecting the availability and vulnerability of different prey species.

It is difficult to measure the influence of prey defensive characteristics on predator behaviour. In open habitats, many ungulate species form herds, which can reduce prey vulnerability. Taylor [97] suggested herd formation, almost always benefits prey while hinders the predator. However, larger group sizes do not necessarily translate into predator detection probability increases. Contextually, Orsdol [98] reported that lions hunting at night were more significantly detected by conch (*Kobus kob*) than by topi (*Damaliscus korrigum*) despite average conch group size is half that of topi, even if according to Schaller [81], topies are the most vigilant and least vulnerable species in the Serengeti.

Prey may also have other defence tactics. Non-human primates climb trees, wild boars and armadillos refuge in burrows, while others defend themselves with horns, antlers, tusks, and spines. Serious injuries and deaths associated with prey capture have been observed in lions, cougars, and tigers [85,99] but no data were available to indicate the frequency of injuries by prey type. Any injury that incapacitates a lone predator can have serious consequences in most landscapes, except in per-urban settings where predators can find food sources in effect provided for by humans. Prey can also alter their distribution in response to the presence of predators. Temporary shelters can provide protection for prey at certain times of the day, year, or season. For example, chitals would concentrate in pastures near human facilities at dusk, not to graze (as there is little forage), but to protect themselves from predators such as tigers and leopards, which are nocturnal and do not come close anthropogenic areas in many Indian national parks and reserves [100,101].

### 4.3. Natural Physical Environment Dependent Factors

Hunting success in all cat species has been reported to depend entirely on the physical characteristics of environment in which predatory interaction occurs. Cats use almost any surface or cover to get as close to the prey as possible before the last attack. In this context, minimum distance travelled in the last charge has been reported to strongly correlate with successful prey death. Using data from real-life stalking and computer simulations, Elliott et al. [102] showed that Ngorongoro African lions had a high probability (80%) of catching a Thomson gazelle (*Gazella thomsoni*) when they launched the attack at a distance of 7.6 m or less, while the chances of success in hunting wildebeests and larger zebras reduced to 50% at distances of 15.2 m. Our study suggested lake and mountain environments to be those for which the highest probability of prey successful escape was reported. Such contexts may difficult chasing phase as cats are rarely habituated to aquatic environments, except for the jaguar or to mountainous areas, such as snow leopard. Therefore, open habitats such as grasslands and forest habitats increase the probability of success in hunting, even if they lack obstacles, and could presumably favour prey escape [68].

Vegetation cover provides prey with a camouflage element [101]. Grass cover has proven to directly influence the distance covered in the final chase during the stalking phase. Elliott et al. [102], Orsdol [98] and Schaller [81] suggested lions hunting on 0.3 to 0.6 m grass covered fields had twice as much success as those hunting on less than 0.3 m grass covers, while hunting success boosted on 0.8 m grass covered fields.

Since most cats are nocturnal [103], they hunt in the dark, hence it may be logical to assume that lunar phase could also affect hunt success. However, logistical problems have hindered most field scientists from studying this factor. Contextually, as research by Emmons et al. [104] suggested, even if southeastern Peru ocelots spend the same time amount hunting on either dark or moonlit nights, their hunting limited to dense, covered areas on dark nights. This was supported by Orsdol [98] who reported lunar phase to influence hunting success in lions who were twice as successful in capturing prey on dark nights.

Climatic factors can limit feeding time and even affect prey susceptibility or predator effectiveness. Hornocker [85] found that cougars were able to kill more elk in winter when deep snow forced them to head for steep terrain where they were more vulnerable. In Nepal, tigers hunted less during the day in the hot season while during flood periods they shifted their activities to higher, drier grounds [89]. In this context, Orsdol [98] noted that African lions tended to start hunting more often when storms were imminent and reported a hypothetic reduced ability of preys to detect predator under such conditions.

The theoretical knowledge of these conditioning elements in the predatory behaviour of big cats will allow the reinforced approach of selection strategies potentially extrapolated to domestic animals. At the same time, it will seek to promote the conservation and transmissibility of desirable behaviour patterns in domestic feline species that can act as predators (for example, the cat) or as prey (domestic livestock/wildlife conflicts).

Two different selective proposals arise in what in biology is known as evolutionary arms race [105]. An evolutionary arms race is an evolutionary competence between sets of coevolving genes that develop reciprocal adaptations and counter-adaptations in different species, as in an evolutionary arms race between a predator species and its prey [106]. The first strategy would be to select for better predators. For instance, in the case of domestic cats, for the control of pest which cause great losses in agricultural farms. The second strategy would be the selection for more maternal and smarter ruminants more easily defending or escaping from big cats which is troublesome for shepherds in rural livelihoods in South America and Africa and one of the main detriments for big cat in situ conservation. Such strategies may be supported as follows.

On the one hand, cats took over the role as pest controllers to the extent that humans began to appreciate their presence and spread them worldwide, along with their cereals and mice. Unlike dogs, however, historically modelled by breeding and the need to fit into human societies [107], cats did not change that much compared with their wild ancestors. Indeed, genetic analyses confirm that, for the longest part of the shared history of humans and cats, the deal was one of mutual tolerance rather than active domestication. As a piece of evidence, early signs of selection in cats may appear much later than in dogs and are mainly linked to the diversification of cats into the wide variability of worldwide breeds which started in Europe from the Middle Ages onwards [108].

Rat infestations regulation ability of domestic cats is usually attributed to the more efficient behaviour of some cats as rat catchers, either by constitution or training. This translates into cats being considered from pest controllers in farms to mere ‘companion’ animals [109]. This wide variability enables the implementation of selection strategies for additive traits. For selection to be efficient, studying the environmental factors involved in the success of either prey escape or predator hunt is necessary [110]. For instance, young cats have been reported to hunt more diverse prey than older more experienced ones from whom humans especially benefit, as they are rodent controllers in rural livelihoods [111]. Literature has suggested keeping “super predator” cats in houses, especially in urban areas, prevents rodent predation on birds [111]. Additionally, new biologically friendly strategies focus on the use of wild feline predators, such as leopard cats, as biological pest-controller of rats in oil palm plantation landscapes as effective replacers of traditionally used expensive and environmentally polluting chemical rat poisons [112,113].

On the other hand, big cat/human conflicts (over depredation on small and large livestock), is one of the major causes of large carnivore species global decline. The likelihood of such conflicts to occur increases as big cats often shift from natural to livestock prey due to an increased proximity to agriculture along tropical deforestation frontiers [114]. Contextually, livestock husbandry significantly affected depredation rates, hence killed predators’ numbers [115]. According to Ogada et al. [115], traditional, low-tech husbandry approaches can make an important contribution to the conservation of big cats through the reduction of the incidence of such conflicts. For instance, some livestock species such as buffalo display defensive behaviour against predators due to their increased maternal behaviour. Indeed, livestock mortality associated with big cats (jaguar and puma) may be reduced by keeping buffaloes and cattle in the same paddock, only buffaloes [116] or by selecting cattle with enhanced maternal behaviours.

Selecting for stronger maternal behaviours in ruminants to suit low-labour input systems is emerging. However, aggressive defence of the offspring can result in serious and fatal injuries to producers, veterinarians, carers or others as a side effect, thus selection and factors involved (such as age or type) must be approached carefully. Bearing this in mind, selecting individuals with an intermediate expression of fear and maternal care or including negative selection against aggressiveness in breeding programmes may benefit handler safety while protection of offspring against predators may still be ensured [117].

## 5. Conclusions

The unique characteristics of each animal species conditions the different hunting techniques that they implement. The younger the animal, the greater the probability of failure during the hunt, mainly due to its relatively smaller body size and practical inexperience compared to adult individuals of its species and population. Adult animals, with notable practical experience and healthy, are more effective during the hunting tasks, while their capacity of resistance against the defence exercised by the prey will be greater and the probability of regretting outstanding physical injuries is reduced. The longer the time between attention and first contact the more likely the prey was to finally manage to escape. Larger preys make it more difficult for the predator to efficiently access body locations on which to exert its immobilization and final attack. The relief of the terrain is the only environmental factor that affects the success of big cat hunting. Aquatic environments of considerable extension (lake) and mountainous environments, make the tasks of stalking and pursuit of prey difficult, as these habitats provide them with opportunities to avoid the attack of predators, as the latter are fundamentally adapted to terrestrial environments with relative flatness.

## Figures and Tables

**Figure 1 animals-12-00051-f001:**
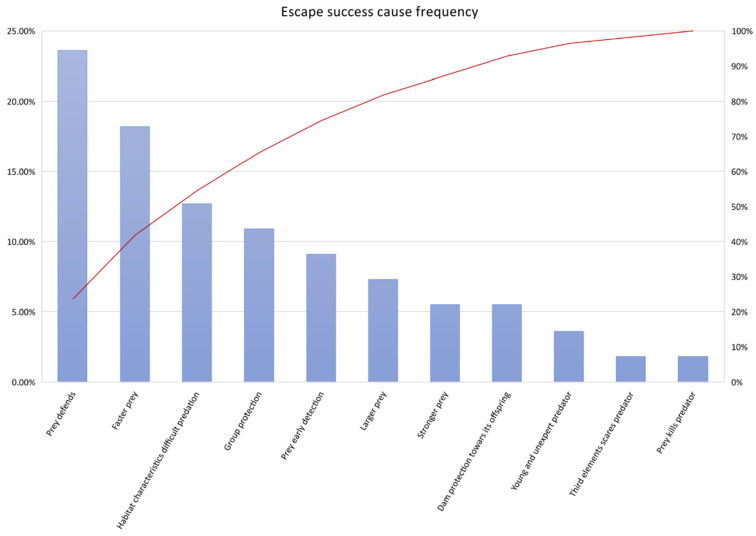
Summary of frequency analysis results for causes of successful escape in large cat prey.

**Table 1 animals-12-00051-t001:** Summary of results from Bayesian ANOVA to detect differences in mean probability of success in prey escape across predator-related factors and covariates.

Factor/Covariate	Groups	Sum of Squares	Degrees of Freedom	Medium Square	F	*p*-Value	Bayes’ Factor
Species	Between	4.177	10	0.418	2.525	0.007	0.000
	Within	38.374	232	0.165			
Sex	Between	0.230	1	0.230	1.393	0.241	0.152
	Within	17.204	104	0.165			
Age	Between	0.764	1	0.764	4.405	0.037	0.448
	Within	41.788	241	0.173			
Type of attack	Between	0.466	1	0.466	2.670	0.104	0.191
	Within	42.085	241	0.175			
Hunting mode	Between	0.428	2	0.214	1.220	0.297	0.014
	Within	42.123	240	0.176			
Number of predators	Between	2.220	10	0.222	1.277	0.244	0.000
	Within	40.331	232	0.174			
Time from first attention to action (seconds)	Between	59.259	1	59.259	0.129	0.721	0.114
	Within	23,944.000	52	460.462			
Time from first attention to interaction or direct contact with the prey (seconds)	Between	3507.379	1	3507.379	5.130	0.028	1.223
	Within	32,136.580	47	683.757			
Contact point	Between	0.845	8	0.106	0.887	0.528	0.000
	Within	23.331	196	0.119			
Status of the predator at the end of the hunt ^1^	Between	2.434	2	1.217	7.282	0.001	4.651
	Within	40.117	240	0.167			
Hunting attempts	Between	1.312	3	0.437	2.561	0.056	0.018
	Within	40.639	238	0.171			

^1^ Status of the predator at the end of the hunt described whether the predator was apparently healthy, visually injured or dead after predation interaction. F = Snedecor’s F.

**Table 2 animals-12-00051-t002:** Summary of results from Bayesian ANOVA to detect differences in mean probability of success in prey escape across prey-related factors and covariates.

Factor/Covariate	Groups	Sum of Squares	Degrees of Freedom	Medium Square	F	*p*-Value	Bayes’ Factor
Species	Between	11.958	52	0.230	1.423	0.046	0.000
	Within	30.542	189	0.162			
Number of simultaneous prey	Between	0.000	1	0.000	0.016	0.899	0.058
	Within	0.995	187	0.005			
Sex	Between	0.000	1	0.000	0.002	0.966	0.098
	Within	8.437	62	0.136			
Age	Between	0.187	1	0.187	1.059	0.304	0.087
	Within	42.261	239	0.177			

**Table 3 animals-12-00051-t003:** Summary of results from Bayesian ANOVA to detect differences in mean probability of success in prey escape across environment-related factors and covariates.

Factor/Covariate	Groups	Sum of Squares	Degrees of Freedom	Medium Square	F	*p*-Value	Bayes’ Factor
Time of day	Between	0.050	1	0.050	0.285	0.594	0.059
	Within	42.501	241	0.176			
Atmospheric conditions	Between	0.583	4	0.146	0.827	0.509	0.000
	Within	41.968	238	0.176			
Relief	Between	3.459	11	0.314	1.858	0.046	0.000
	Within	39.093	231	0.169			

**Table 4 animals-12-00051-t004:** Summary of the descriptive statistics of the posterior distribution for the predator species factor.

Parameters	Posterior			95% Confidence Interval	
Levels	Mode	Mean	Variance	Lower Limit	Upper Limit
Tiger	1.077	1.077	0.004	0.949	1.205
Leopard	1.222	1.222	0.004	1.103	1.342
Snow Leopard	1.714	1.714	0.024	1.411	2.017
Melanistic Leopard	1.000	1.000	0.083	0.433	1.567
Puma	1.538	1.538	0.013	1.316	1.761
Jaguar	1.273	1.273	0.015	1.031	1.514
Cheetah	1.256	1.256	0.004	1.134	1.378
Leon	1.200	1.200	0.002	1.110	1.290
Serval	1.000	1.000	0.167	0.199	1.801
Ocelot	1.000	1.000	0.167	0.199	1.801
Caracal	1.000	1.000	0.167	0.199	1.801

**Table 5 animals-12-00051-t005:** Summary of descriptive statistics of the posterior distribution for the predator’s age range factor.

Parameters	Posterior			Posterior	
Levels	Mode	Mean	Variance	Lower Limit	Upper Limit
Young	1.438	1.438	0.011	1.232	1.643
Adult	1.211	1.211	0.001	1.157	1.266

**Table 6 animals-12-00051-t006:** Summary of the descriptive statistics of the posterior distribution for the prey species factor.

Parameters	Posterior			Posterior	
Levels	Mode	Mean	Variance	Lower Limit	Upper Limit
Baboon	1.000	1.000	0.082	0.439	1.561
Bear	1.000	1.000	0.163	0.207	1.793
African Stork	1.000	1.000	0.163	0.207	1.793
Boar	1.000	1.000	0.041	0.604	1.396
Buffalo	1.158	1.158	0.004	1.029	1.287
Capibara	1.200	1.200	0.033	0.845	1.555
Caracal	1.000	1.000	0.163	0.207	1.793
Catfish	1.000	1.000	0.163	0.207	1.793
Crocodile	2.000	2.000	0.163	1.207	2.793
Red deer	1.095	1.095	0.008	0.922	1.268
Donkey	1.000	1.000	0.163	0.207	1.793
Duiker	1.000	1.000	0.163	0.207	1.793
African Elephant	1.400	1.400	0.033	1.045	1.755
Fennec	2.000	2.000	0.163	1.207	2.793
Giant Otter	2.000	2.000	0.163	1.207	2.793
Giraffe	1.333	1.333	0.054	0.876	1.791
Guanaco	1.556	1.556	0.018	1.291	1.820
Guinea fowl	1.500	1.500	0.082	0.939	2.061
Hare	1.000	1.000	0.082	0.439	1.561
Himalayan Ibex	1.667	1.667	0.054	1.209	2.124
Hippopotamus	1.500	1.500	0.082	0.939	2.061
Impala	1.111	1.111	0.009	0.924	1.298
Jackal	1.000	1.000	0.163	0.207	1.793
Kudu	1.000	1.000	0.041	0.604	1.396
Markhor	2.000	2.000	0.082	1.439	2.561
Rhesus macaque	1.000	1.000	0.054	0.542	1.458
Pyrenean ibex	2.000	2.000	0.163	1.207	2.793
Mule deer	2.000	2.000	0.163	1.207	2.793
Oryx	2.000	2.000	0.163	1.207	2.793
Ostrich	1.000	1.000	0.041	0.604	1.396
Owl	1.000	1.000	0.163	0.207	1.793
Reedbuck	1.000	1.000	0.163	0.207	1.793
Rhinoceros	1.000	1.000	0.163	0.207	1.793
Roan antelope	1.000	1.000	0.163	0.207	1.793
Sloth	1.000	1.000	0.163	0.207	1.793
Springbok	1.000	1.000	0.082	0.439	1.561
Steenbok	2.000	2.000	0.163	1.207	2.793
Topi	1.000	1.000	0.163	0.207	1.793
Warthog	1.333	1.333	0.007	1.171	1.495
Wildebeest	1.190	1.190	0.008	1.017	1.364
Yak	2.000	2.000	0.163	1.207	2.793
Zebra	1.417	1.417	0.014	1.188	1.646
Zebu	1.000	1.000	0.082	0.439	1.561

**Table 7 animals-12-00051-t007:** Summary of the descriptive statistics of the posterior distribution for the factor status of the predator at the end of the hunting interaction.

Parameters	Posterior			95% Confidence Interval	
Levels	Mode	Mean	Variance	Lower Limit	Upper Limit
Apparently healthy	1.213	1.213	0.001	1.161	1.265
Visually injured	2.000	2.000	0.056	1.535	2.465
Dead	2.000	2.000	0.169	1.195	2.805

**Table 8 animals-12-00051-t008:** Summary of descriptive statistics of the posterior distribution for the relief factor.

Parameters	Posterior			95% Confidence Interval	
Levels	Mode	Mean	Variance	Lower Limit	Upper Limit
Snow-covered	1.000	1.000	0.057	0.532	1.468
Pond	1.125	1.125	0.021	0.838	1.412
Savannah	1.218	1.218	0.001	1.157	1.278
Forest	1.059	1.059	0.010	0.862	1.255
Rural road	1.000	1.000	0.171	0.189	1.811
Rocky terrain	1.000	1.000	0.171	0.189	1.811
Lake	1.500	1.500	0.085	0.927	2.073
Mountain	1.667	1.667	0.014	1.433	1.901
River	1.286	1.286	0.012	1.069	1.502
Jungle	1.000	1.000	0.171	0.189	1.811
Highway	1.250	1.250	0.043	0.845	1.655

**Table 9 animals-12-00051-t009:** Summary of descriptive statistics of the posterior distribution for the time from first attention to interaction or direct contact with the prey (seconds) covariate across hunting interaction success possibilities.

Parameters	Posterior 95% Confidence Interval				
Levels	Mode	Mean	Variance	Lower Limit	Upper Limit
Unsuccessful escape	39.043	39.043	15.525	31.287	46.800
Successful escape	74.333	74.333	238.049	43.962	104.705

**Table 10 animals-12-00051-t010:** Summary of results for the Bayesian ANOVA approach for the model predicting the probability of successful dam escape including factors for which a significant effect had been detected.

Source	Sum of Squares	Degrees of Freedom	Medium Square	F	*p*-Value
Regression	15.526	69	0.225	1.435	0.032
Residual	26.974	172	0.157		
Total	42.500	241			
Dependent variable: Probability of escape success.					

Model with independent variables: (Intercept), predator species, age range of the predator, prey species, status of the predator at the end of the hunting interaction, relief.

**Table 11 animals-12-00051-t011:** Summary of the explanatory power of the variability captured by the proposed model for the probability of successful escape.

Bayes’ Factor	R	R Square
0.000	0.604	0.365

## Data Availability

Data will be made available from the authors upon reasonable request.

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
