# Peer review of "The Winner Takes it All: Risk Factors and Bayesian Modelling of the Probability of Success in Escaping from Big Cat Predation"

_animals, 2021, doi:10.3390/ani12010051_

Round 1
Reviewer 1 Report
The current manuscript presents some interesting, but ultimately intuitive findings about factors influencing the success of prey acquisition by big cats. Much of the findings of the current study are already born out in detailed published studies and the authors need to make clear how their findings value add to that already known.
I have attached an annotated version of the manuscript with many comments and suggestions for improving the text. Rather than itemise them here I would encourage the authors to read them and respond in full with appropriate changes. Most suggested edits relate to improving English expression, but there are several comments relating to clarifying results or putting them in better context. This includes putting findings in perspective, in relation to original aims or with existing literature.

Author Response
Reviewer 1
The current manuscript presents some interesting, but ultimately intuitive findings about factors influencing the success of prey acquisition by big cats. Much of the findings of the current study are already born out in detailed published studies and the authors need to make clear how their findings value add to that already known.
I have attached an annotated version of the manuscript with many comments and suggestions for improving the text. Rather than itemise them here I would encourage the authors to read them and respond in full with appropriate changes. Most suggested edits relate to improving English expression, but there are several comments relating to clarifying results or putting them in better context. This includes putting findings in perspective, in relation to original aims or with existing literature.
Response: We thank the reviewer for his/her suggestions and appreciate the time and effort made. All changes proposed have been applied as we feel they really improve the quality of our paper. A point-by-point response to comments is provided as well as a file where changes are highlighted.
Delete "to".
Response: Deleted.
You need to stipulate here that you obtained materials from videos post on-line and which were globally available. In simple terms, spell out the source of your "data".
Response: Added.
The results of this work seemingly have many potential applications, but was THE driving force behind doing the research? Put simply, what was the most important reason?
Response: Added and rewritten.
"escaping"
Response: Changed.
Combinations
Response: Changed.
"The most parsimonious model ...."
Response: Added.
"escaping"
Response: Changed.
Insert "on wild prey populations".
Response: Added.
Delete "on wild prey populations" here and transfer to earlier in the sentence.
Response: Deleted.
Delete comma.
Response: Deleted.
Delete comma.
Response: Deleted.
"regard" (singular)
Response: Changed.
"regulating" rather than "controlling"? Many other factors help influence the abundance of prey species.
Response: Changed.
"mesopredators".
Response: Changed.
"Mesopredators"
Response: Changed.
Delete "that they enjoy today." Unsure whether they actually enjoy it, but rather an innate instinct for survival.
Response: Deleted.
Include references to these "many scholars" please.
Response: References were added.
Replace with "medium-sized"
Response: Replaced.
Insert "and immobilise it."
Response: Inserted.
"high"?
Response: Changed.
Insert "ordinarily".
Response: Inserted.
"the snow leopard (Panthera uncia)"
Response: Changed.
This sentence is overly wordy. How about "successful predation"?
Response: We agree and changed it.
Insert "approach".
Response: Inserted.
Replace with "vegetation".
Response: Replaced.
You need to state somewhere in this paragraph that you utilised video footage of predator/prey interactions obtained from various web sources.
Response: Added.
This sentence is way too long. Please break up into at least two sentences, retaining existing content though.
Response: We rewrote the paragraph to make it more easily readable.
Delete additional space.
Response: Deleted.
Was this the total "population" of videos that were available or a subset of a larger number, the residual of which was deemed not suitable?
Response: This was explained.
This sentence needs re-organising to make clearer. Something like: "Under normal circumstances it would have been prudent to collect our own data from a field study to examine the phenomenon of predation by big cats. However, as the COVID-19 pandemic situation effectively precluded this, we instead sought information from existing studies ...." Associated with this, think about placing this text at the beginning of the methods section, as it sets the context for the approach taken. Then get into the bit about how you sought out videos and then
compartmentalised animal behaviours.
Response: The whole Section was organized to make it more readable.
Should be "32.80%"
Response: Corrected.
Insert "the"
Response: Inserted.
"relationship" rather than "basis".
Response: Changed.
Delete additional space.
Response: Deleted.
Add a space here.
Response: Added.
Insert "variously".
Response: Inserted.
"visualised"?
Response: Changed.
Delete "Rather than attacking fails" and start the sentence with "The defensive mechanisms implemented by prey ...."
Response: Rewritten.
Insert "of the".
Response: Inserted.
Delete "a".
Response: Deleted.
Delete "dam".
Response: Deleted.
Include an introductory sentence here that reiterates the context behind the study, before launching into the results.
Response: Added.
But does this account for the small sample sizes for these latter species? Maybe if you had more footage of them you might witness more successful predation events?
Response: Indeed, this is the particular reason for which a Bayesian approach. To be able to sort potential distorting conditions derived from sampling such as unequal representation of observations for each predator in the study sample, Bayesian estimation methods were used, as suggested by other authors [54]. These methods have been reported to require a much smaller ratio of parameters to observations (1:3 instead of 1:5); that is, Bayesian inference maximizes the ability to determine significant effects for relatively limited sample sizes.
These sample limitations are not cost-free but are reflected in the broadening of confidence intervals, which must be accompanied by an acceptable Bayes factor value. Furthermore, there are qualities of the Jeffrey–Zellner–Siow (JZS) prior that make it especially appropriate when performing linear regression analyses. Among these, the prior is symmetric and centered at zero in line with the predictive matching criterion, hence positive and negative values of the slope parameters have a priori the same probability to occur. Furthermore, JZS prior is scale invariant, thus the resulting Bayes factor does not depend on the scale of both the dependent variable and factors or covariates, hence results do not change when different unit variables are evaluated together, which is common in field conditions studies [54].
Change to "selectively stop hunting".
Response: Corrected.
Reword to something like: "Predation success differs with age of the predator, with older animals more likely to capture prey than young animals."
Response: Reworded.
The other point to make here is that social cooperation in hunting may be more important in certain settings and for certain prey species - include this in the summary as well.
Response: Added.
Delete additional space.
Response: Deleted.
Delete additional space.
Response: Deleted.
Replace "rest" with "range of".
Response: Replaced.
Delete "the" and start sentence with "Optimum".
Response: Deleted.
You need to put the subsequent set of paragraphs into context with the main findings from your study. At the moment it reads like a description of the main prey of different big cats. But how does this relate to your findings? Please put in better context.
Response: Added.
", even if the latter were more abundant." (i.e. join the two sentences).
Response: Joined.
"deer" (i.e. singular)
Response: Changed.
"prey on sambar deer"
Response: Changed.
Delete "study".
Response: Deleted.
Fairer to say that will be the case in most landscapes, except in per-urban settings where predators can find food sources in effect provided for by humans.
Response: Reworded.
This paragraph provides insights into your study findings and so should be elevated above the text where you have simply described the main prey items of the different predators.
Response: Moved to the section suggested by reviewer.
Replace "is" with "may be".
Response: Changed.
Insert "of prey".
Response: Inserted.
"prey" (i.e. singular)
Response: Changed.
Reviewer 2 Report
This paper represents a nice effort to document rather frequent episodes of hunting for the animals though very difficult to observe and document from the beginning to the end of the episode. The reliablity of the variables in the table 1, and the complete absence of other indications about sample question al the results of this paper. These information appear later in the paper (line 178-186). So this paragraph should appear before in that section.
The distance between the predator and its prey at the time of first attention is never mentioned. This might be an important factor for success -or failure- of the predation.
The style is often wordy and irrelevant. The tables are very difficult to read for a basic reader, as I am.
The sophisticated statistics on variables that often seem to be difficult to be reliable contrast with the largely descriptive aspect of this paper. The statistics appear to be disconnected from the very essence of the paper.
Some references appear to be highly biased. It is not fair to cite recent papers for items related to definiton of predation and predator/prey equilibrium. This is something that has been studied for years before now, so it would be better to cite previous 'old' papers or use something like « e.g. » for the most recent papers.
Line 98 « on the other hand » implies that it exists a « on one hand » before, in the previous sentence.
Line 106 this assumption needs a reference
Line 111 the reference should be replaced by its rank in the reference list
line 128 « victorious clumination ... ». Loose, wordy style
line 134 « cat herd » only applied to the very few large felids living in groups, e.g. Lion.
Line 137 and previously « venatorial ». I must admit that I don't know this word. Is there a simpler expression to express the same meaning ?
Line 142-143 So every species ! Meaning not clear.
Line 171. the citation of table 1 is quite bizarre and seems unrelated to the text.
Line 178-186 It is dubious that the COVID-19 situation has prevented the authors to move around at least 4 continents to videotape episodes of predation on large felids, as assumed lines 167-168. So these lines might be deleted.
Author Response
Reviewer 2
This paper represents a nice effort to document rather frequent episodes of hunting for the animals though very difficult to observe and document from the beginning to the end of the episode. The reliablity of the variables in the table 1, and the complete absence of other indications about sample question al the results of this paper. This information appear later in the paper (line 178-186). So this paragraph should appear before in that section.
Response: We thank the reviewer for his/her suggestions and appreciate the time and effort made. We modified the whole section to make it more readable and chronologically logical. We understand the concerns stated by the reviewer, but we extracted the only information from videos which could be reliably taken, after a careful selection of unedited videos, which could have distorted our results. Indeed from the complete database of 308 videos only 244 remained.
The distance between the predator and its prey at the time of first attention is never mentioned. This might be an important factor for success -or failure- of the predation.
Response: We agree. However, this could not be measured from the video, hence, this was the reason for us not to consider it in the analyses. Indeed, this and other potentially excluded variables or factors are represented by the error term of the equations. If this (and others) had been registered explanatory potential of the model would have been higher (although it is already acceptable) and error would have been lower (although it is acceptable).
The style is often wordy and irrelevant.
Response: We agree and rewrote the sections following the reviewer suggestion to make it more easily readable and to correct typos and grammar inconsistencies. A Cambridge ESOL examination instructor revised the manuscript to ensure potential incidences were solved.
The tables are very difficult to read for a basic reader, as I am.
Response: We understand the reviewer concern. Bayesian inference is complex to those who are not familiar. However, what we present here are descriptive statistics and ANOVA results, which are presented as it can be usually found in literature for similar results.
The sophisticated statistics on variables that often seem to be difficult to be reliable contrast with the largely descriptive aspect of this paper.
The statistics appear to be disconnected from the very essence of the paper.
Response: Our study is partially descriptive. However, other techniques such as Bayesian inference for ANOVA and linear regression were used. Indeed, this is the particular reason for which a Bayesian approach (as it may be the only reliable approach in the context of this study). To be able to sort potential distorting conditions derived from sampling. Bayesian estimation methods were used, as suggested by other authors [54]. These methods have been reported to require a much smaller ratio of parameters to observations (1:3 instead of 1:5); that is, Bayesian inference maximizes the ability to determine significant effects for relatively limited sample sizes.
These sample limitations are not cost-free but are reflected in the broadening of confidence intervals, which must be accompanied by an acceptable Bayes factor value. Furthermore, there are qualities of the Jeffrey–Zellner–Siow (JZS) prior that make it especially appropriate when performing linear regression analyses. Among these, the prior is symmetric and centered at zero in line with the predictive matching criterion, hence positive and negative values of the slope parameters have a priori the same probability to occur. Furthermore, JZS prior is scale invariant, thus the resulting Bayes factor does not depend on the scale of both the dependent variable and factors or covariates, hence results do not change when different unit variables are evaluated together, which is common in field conditions studies [54]. Furthermore, confidence intervals are provided for every parameter which to ensure a measure of the reliability of each of them is reported.
Some references appear to be highly biased. It is not fair to cite recent papers for items related to definiton of predation and predator/prey equilibrium. This is something that has been studied for years before now, so it would be better to cite previous 'old' papers or use something like « e.g. » for the most recent papers.
Response: We tried to keep the introduction and discussion updated. Although we agree with the reviewer in that certain concepts were already defined decades ago. We included an old reference as we do think this is important to credit as well. Still adding new references helps highlighting the conservation of the terms along time.
Line 98 « on the other hand » implies that it exists a « on one hand » before, in the previous sentence.
Response: We agree and changed it.
Line 106 this assumption needs a reference
Response: References were added.
Line 111 the reference should be replaced by its rank in the reference list
Response: Suggestion was followed.
line 128 « victorious clumination ... ». Loose, wordy style
Response: We simplified.
line 134 « cat herd » only applied to the very few large felids living in groups, e.g. Lion.
Response: Clarified.
Line 137 and previously « venatorial ». I must admit that I don't know this word. Is there a simpler expression to express the same meaning ?
Response: Venatorial means related to hunt or hunting. We changed this along the body text.
Line 142-143 So every species ! Meaning not clear.
Response: We clarified this in the body text.
Line 171. the citation of table 1 is quite bizarre and seems unrelated to the text.
Response: We understand the reviewer concern and rewrote the section.
Line 178-186 It is dubious that the COVID-19 situation has prevented the authors to move around at least 4 continents to videotape episodes of predation on large felids, as assumed lines 167-168. So these lines might be deleted.
Response: We deleted them.
Round 2
Reviewer 2 Report
The authors made a valuable effort to answer all the concerns expressed in the first review. However this paper remains problematic on many grounds. I am still incompetent to judge the use of the Bayesian statistics. However I am able to see that the discussion represents 45% of this paper. This is much too large and leads to the impression that this paper is mostly descriptive and that the statistic effort is useless. Many style flaws remain. They are mentioned in the text. The commented file is attached to this form.

Author Response
Reviewer 2
The authors made a valuable effort to answer all the concerns expressed in the first review. However this paper remains problematic on many grounds. I am still incompetent to judge the use of the Bayesian statistics. However I am able to see that the discussion represents 45% of this paper. This is much too large and leads to the impression that this paper is mostly descriptive and that the statistic effort is useless. Many style flaws remain. They are mentioned in the text. The commented file is attached to this form.
Response: We appreciate the work and effort of the reviewer on revising our paper a second time. A point-by-point response to comments is provided as well as a file where changes are highlighted.
Delete globally available
Response: Deleted.
Big cats do not show anti-predation defense. They show or have to show srtategies to fend off anti-predation strategies of the prey. Change
Response: Changed.
Not clear! In providing preys and enough space to express their strategies. Be more precise
Response: Changed.
Also not clear. Your plans seem to change an equilibirum with better predators and smarter preys. What is the rationale of that. Domestic predators have been selected against predation (dogs) and/or to feed on something different than preys (cats). For cats the trends is to prevent them to prey on birds as people consider that they are responsible for the decline of birds population. The fate of big cats is different. They are endangered as every wild species to to anthropogenic pressures. So I do not understand the applied aspect of this paper. Please clarify or change.
Response: We disagree, but understand the need for clarification as suggested by the reviewer. Cats and dogs are still used, not marginally but frequently, for pest regulation in many rural livelihoods today. It is true that the plans that we propose here can be connected but the word bidirectional may not be the best choice, as they may not be reciprocal in the same context (that is to model a single relationship between a prey species and its predator species). Considering the aforementioned, what we proposed was two different selective approaches. One strategy would be to select for better predators, for instance, in the case of rater dogs or terriers or domestic cats, for the functional role of pest control which causes great losses in agricultural farms (Please see citations enclosed saying that bird predation is marginal in cats. Blaming them for the decrease in bird population may not be appropriate, given as another paper states they may buffer the effect of mesopredators (rats) on birds) while the other strategy be linked to the selection for maternal or smarter ruminants which may more easily defend or escape from big cats attacks which is troublesome for shepherds in rural livelihoods in South America and Africa and one of the main detriments for big cat in situ conservation. For further clarification, we added a section discussing this in the discussion section.
Elton, C. S. 1953. The use of cats in farm rat control. The British Journal of Animal Behaviour 1(4):151-155. doi: https://doi.org/10.1016/S0950-5601(53)80015-8
Courchamp, F., M. Langlais, and G. Sugihara. 1999. Cats protecting birds: modelling the mesopredator release effect. Journal of Animal Ecology 68(2):282-292.
Gross, M. 2020. Of mice and men, cats and grains. Elsevier.
Kauhala, K., K. Talvitie, and T. Vuorisalo. 2015. Free-ranging house cats in urban and rural areas in the north: useful rodent killers or harmful bird predators? Folia Zoologica 64(1):45-55, 11.
Michalski, F., R. Boulhosa, A. Faria, and C. Peres. 2006. Human–wildlife conflicts in a fragmented Amazonian forest landscape: determinants of large felid depredation on livestock. Animal conservation 9(2):179-188.
Ogada, M. O., R. Woodroffe, N. O. Oguge, and L. G. Frank. 2003. Limiting depredation by African carnivores: the role of livestock husbandry. Conservation biology 17(6):1521-1530.
Silmi, M., M. Mislan, S. Anggara, and B. Dahlen. 2013. Using leopard cats (Prionailurus bengalensis) as biological pest control of rats in a palm oil plantation. Journal of Indonesian Natural History 1(1):31-36.
Improving efficiency to feed on prey IS adaptation!
Response: We deleted the word adaptation.
Is there an orography other than "of the terrain"? It seems like a pleonasm. In addition orography refers specifically to moutains. Are all thre cited predator species living in mountains? The term "Relief" is more general and possibly more relevant.
Response: Changed across the manuscript.
You talk about predatory species and only mention 2 felids species! There is also a number of birds of prey and other species than felids - mustelids-. Change
Response: Changed.
again there is a lot of birds of prey. They do not fit well into your description. So please state clearly that in your paper you are dealing with a very limited number of terrestrial predators. This is not a problem. The problem is to make loose generalities. In addition the predators you are talking about should be suitable for videos! I've been aware of the difficulties to document predation in lions at night (cf "Akagera"). And the lions still chase at night in a suitable savanah environment to follow them. What about forest predators at night? This all the difficutlies to document this important behavior. I then suggest that at the very beginning of your paper you give a broad picture of preadtory behaviors before restricting your viws to what is available for study (as it starts on line 86. However before that your "restricted views" are already dominant.
Response: We followed the reviewer suggestion and provided a broad picture of predatory behaviors before jumping into the case of big cats.
Useless. Delete
Response: Deleted.
No. not complex!
Response: Changed.
Social or gregarious? Wordy. Choose one words. Social species do show a gregarious trait though all greagarious species are not social
Response: Changed.
STYLE! much too emphatic! Not relevant in such a paper.
Response: Deleted.
It seems to me that nowhere you mention the activity rythms of the species and that some species are diurnal - suitable for filming- and others are strictly nocturnal or may feed -chase both durin the day and during the night. This is, to me at least, an important point. This is only incidentally mentionned line 133.
Response: Changed.
What are the other species of "large felids living in groups"? From those cited in this paper, I don't find any others
Response: Changed.
Videos during the day or at night or both?
Response: Both were considered via the time of the day variable.
Not convincing at all. See in the abstract
Response: We rewrote the section.
Do you mean that ALL the videos display success in predation? I see a big bias here! Please clarify
Response: No, this means that success in hunting was searched for, but as well as we searched for fail in hunting. We clarified.
Though it is a big effort and probably highly instructive, to incorporate theses "cases" in a dataset of much larger samples seems to me totally irrelevant and useless
Response: We understand the reviewer concern, but including them still provides information, which in the context of the scarcity of resources dealing with them to be found, is relevant.
This table is in the Methods section instead of being in the Results section
Response: Moved to results.
What is meant by "Status of the predator at the end of the hunt"? This should be defined before the table
Response: We added explanation as a footnote.
What is the difference between "state" and "status"? Status in the table, state in the text. Please homogenize
Response: We homogenized it to status.
This figure is nice as it is easy to understand instead of the lengthy and indigestible tables
Response: Thank you.
Not very surprising! A little trivial unless the older predator is much too old!
Response: We agree, but it is still a finding.
But considering the number of solitary species in large felids and the uniqueness of a certain form of sociality in lions, and the general success in predation for most of the species, this sentence is useless. Cooperation in predation does not seem to be a factor that have driven the sociality trait during evolution. Cooperation in predators for hunting might be an emergent property of sociality. Solitary species seems to be as successful as the lions. By the way to which species belong the "dead predators"?
Response: We removed it. Anyway, it was suggested to be added by the other reviewer at a previous round. Lions were.
This manuscript is a resubmission of an earlier submission. The following is a list of the peer review reports and author responses from that submission.